# Synthesis of Bio-Based Polyurethanes from Functionalized Sunflower Seed Oil

**DOI:** 10.3390/ijms262311380

**Published:** 2025-11-25

**Authors:** Csilla Lakatos, Katalin Czifrák, Csaba Cserháti, Réka Borsi-Gombos, Lajos Nagy, Miklós Zsuga, Sándor Kéki

**Affiliations:** 1Department of Applied Chemistry, University of Debrecen, Egyetem tér 1, H-4032 Debrecen, Hungary; lakatoscsilla@science.unideb.hu (C.L.); czifrak.katalin@science.unideb.hu (K.C.); nagy.lajos@science.unideb.hu (L.N.); zsuga.miklos@science.unideb.hu (M.Z.); 2Department of Solid State Physics, University of Debrecen, Bem tér 18/b, H-4026 Debrecen, Hungary; cserhati.csaba@science.unideb.hu; 3Department of Physical Chemistry, University of Debrecen, Egyetem tér 1, H-4032 Debrecen, Hungary; gombos.reka@science.unideb.hu

**Keywords:** sunflower oil, oil polyol, polyurethane, crosslinking, characterization, scaffold

## Abstract

In this study, bio-based polyurethanes (PUs) were synthesized using renewable polyols derived from sunflower seed oil, aiming to develop flexible yet robust polymeric films and scaffolds. Given their composition and favorable physico-chemical properties, these materials may represent promising candidates for the design and development of advanced biomedical systems. Two distinct oil polyols were prepared via glycerol transesterification (GM) and epoxidation (EPO) with hydrogen peroxide/glacial acetic acid, respectively. These polyols, in combination with poly(tetramethylene ether) glycol (PTMEG) and/or poly(ethylene glycol) (PEG), served as diol components in a one-step reaction with 1,6-hexamethylene diisocyanate (HDI). The structure of the polyol precursors was thoroughly characterized by MALDI-TOF MS and NMR spectroscopy, confirming successful functionalization. The resulting PU films exhibited excellent flexibility (885%) and mechanical properties (23 MPa), as evaluated by ATR-FTIR, Tensile test, DSC, DMA and SEM methods. The crosslink density of the order of 10^−3^ also contributes to the development of outstanding mechanical properties. Stress relaxation experiments were described using a stretched exponential (Kohlrausch–Williams–Watts) model to capture the viscoelastic behavior of the materials. In addition, stress vs. relative elongation curves revealing strain-hardening behavior were also analyzed and modeled mathematically to better describe the mechanical response under deformation. Furthermore, salt leaching techniques were employed to fabricate porous scaffolds. This work highlights the versatility of vegetable oil-based feedstocks in producing functional polyurethanes with tunable mechanical properties for applied polymer systems.

## 1. Introduction

Polyurethanes are synthesized from polyols and isocyanates through the formation of urethane linkages. The relative ratio and molecular structure of these components determine the properties of the resulting polymer [1]. Traditionally, polyurethane production has relied predominantly on petroleum-based raw materials, thereby contributing to the increased anthropogenic environmental impact. But in recent years, the incorporation of bio-based polyols into polyurethane systems [2] has gained significant attention as a sustainable alternative in response to the growing environmental concerns associated with conventional plastics [3]. Bio-polyols serve as partial or full replacements for traditional petrochemical-derived polyols, allowing for the tailoring of polyurethane properties, including mechanical performance, biocompatibility, and biodegradability, which are critical for expanding their potential applications.

The presence of reactive functional groups or unsaturated bonds is a key characteristic of bio-polyols that determines their reactivity and functionality. Among the bio-based feedstocks, vegetable oils stand out due to their abundance, renewability, and versatility of applications ranging from agrochemicals to lubricants and plasticizers [4]. Oils, like sunflower oil are particularly valuable, owing to their high content of unsaturated fatty acids, which introduce reactive double bonds into the molecular structure [5,6].

Polyurethane elastomers are phase-separated materials consisting of alternating soft and hard segments. The soft segments, typically diols, including bio-based polyols, impart elasticity, whereas the hard segments, comprising diisocyanates and short diols or chain extenders, provide mechanical strength [7,8,9,10,11]. Notably, the multifunctionality of certain oil-derived polyols allows them to act as crosslinkers, thereby enhancing the overall mechanical properties of the resulting materials [11].

Chemical modifications such as glycerolysis, epoxidation, and subsequent ring-opening reactions are commonly employed to convert the double bonds of bio-based oils into hydroxyl functionalities. Thus, a variety of reagents and reaction conditions have been developed to optimize these transformations, enabling efficient synthesis of polyols capable of participating in polyurethane formation [12,13,14,15,16,17,18,19,20,21,22].

Furthermore, the combination of modified vegetable oils such as epoxidized or hydroxylated sunflower oil and diisocyanates with polytetramethylene ether glycol (PTMEG) represents a promising approach in the development of advanced polyurethane materials with enhanced sustainability, flexibility, and performance. PTMEG also known as polytetrahydrofuran (PTHF) produced from tetrahydrofurane (THF) that can also be derived from renewable, biomass-based feedstocks in addition to its conventional petrochemical synthesis [23].

This polyurethane system integrates the advantageous properties of the vegetable oil–based polyol and PTMEG, achieving a balance between mechanical strength and elasticity. In contrast, polyurethanes synthesized solely from simple vegetable oils (such as castor, linseed, or soybean oil) and diisocyanates (isophorone diisocyanate, IPDI, or methylene diphenyl diisocyanate, MDI) exhibit significantly lower elasticity [5,22].

This study explores the use of oil-based polyols in elevated proportions to develop polyurethane materials with improved mechanical properties. We specifically investigate how the incorporation of such bio-polyols influences the structural, mechanical and thermomechanical properties of polyurethane networks formed with various building blocks. The insights gained may pave the way for the development of next-generation polyurethanes with bio-based content, suitable for advanced applications in fields such as textiles and biomedical engineering.

## 2. Results and Discussion

### 2.1. Chemical Transformation of Sunflower Oil

To utilize sunflower oil as a polyol precursor in polyurethane synthesis, it must first be functionalized to introduce reactive hydroxyl and/or epoxy groups. The starting sunflower oil (SFO) primarily consists of a mixture of glycerol esters with palmitic acid, oleic acid, and linoleic acid. The double bonds present in the unsaturated fatty acid chains, such as those in oleic and linoleic acids, provide reactive sites for epoxidation. The MALDI-TOF MS spectrum of SFO (Appendix A) confirms a substantial presence of oleic and linoleic acid-derived esters. Assuming equal ionization efficiencies and no mass discrimination effect among the ester species, the relative molar ratios of the different glycerides can be estimated. These calculated values are presented in Figure 1 as a function of the composition.

As shown in Figure 1, the most abundant species in the SFO sample is a glycerol ester containing one oleic acid and two linoleic acids. Based on the relative molar ratios, the average number of double bonds per glyceride molecule in SFO is approximately four (n _=_/n_gl_ ≈ 4). This value is consistent with the result obtained from ^1^H-NMR analysis, where integration of the signals at 5.25 and 5.34 ppm yields an average of approximately 4.2 double bonds per glyceride (n _=_/n_gl_ ≈ 4.2).

The glycerolysis and epoxidation are summarized in Figure 1.

Glycerolysis, or transesterification, was carried out in the presence of excess glycerol under conditions well known in the literature [8] and resulted in an equilibrium mixture of mono-, di-and triglycerides [24,25].

The MALDI-TOF-MS spectrum of the reaction mixture obtained after epoxidation and peracid oxidation is shown in Figure 2.

The full assignment of the MALDI-TOF MS spectrum shown in Figure 2 is provided in Appendix A. In addition to the formation of oxirane rings via epoxidation, the spectrum also indicates partial ring opening, likely due to the addition of water, resulting in the formation of diol structures with two hydroxyl groups. Assuming equal ionization probabilities for all derivatives, it is estimated that approximately 73% of the initial double bonds have reacted. This value is consistent with the extent of conversion calculated from the ^1^H-NMR data, based on the integrals of the signals at 5.25 and 5.34 ppm, which suggest an extent of reaction of about 78%.

Epoxidation was carried out using peracid oxidation. The oxidizing agent was prepared from a mixture of glacial acetic acid and hydrogen peroxide, 1.5 equivalents of hydrogen peroxide were used and the reaction was allowed to proceed for 5 h. When higher amounts of hydrogen peroxide and longer reaction times were applied, triglyceride decomposition was observed. Conversely, shorter reaction times led to only limited conversion. However, under such optimal conditions, neither the epoxidation nor the subsequent ring-opening reactions proceeded to completion, as confirmed by NMR and mass spectrometric investigations.

Furthermore, NMR measurements were also used to confirm the structure of the epoxidized oil (EPO). The ^1^H-NMR spectrum and its assignment and ^13^C-NMR spectrum of EPO are shown in Figure 3 and Appendix A [20,26], respectively while its MALDI-TOF MS spectrum is presented in Appendix A.

In Figure 3, the signals observed at 3.13 and 3.06 ppm correspond to hydroxyl groups formed at the positions of the former double bonds, while the signals between 2.93–2.89 ppm are attributed to the protons of epoxy groups. The protons of unreacted double bonds appear between 5.38–5.24 ppm.

The temperature applied during the epoxidation process can accelerate the decomposition of hydrogen peroxide [27,28] thereby influencing the effective concentration of peracetic acid available for the reaction. As a result, epoxidation may remain incomplete. The process is also accompanied by partial ring-opening of the epoxide groups, which is facilitated by the presence of water in the hydrogen peroxide and glacial acetic acid. However, this reaction remains incomplete as well, since glacial acetic acid is not sufficiently strong to open all the formed epoxide rings.

### 2.2. Synthesis of Polyurethanes

Polyurethane synthesis was carried out in a single step by reacting a mixture of polyols (PTMEG and/or PEG, GM or EPO) with 2 equivalents of 1,6-hexamethylene diisocyanate. The compositions of the reaction mixtures are listed in Table 1. The single-step method was chosen over the two-step approach because, in the latter, the oil-based polyol also acts as a chain terminator, reducing the degree of crosslinking. As a result, uniform polymer films were not consistently obtained with the two-step process.

In the resulting polymer, short-chain and long-chain isocyanate-terminated prepolymers are connected by isocyanate units, along with the oil-based polyol to form a crosslinked network structure (Figure 2). Within this structure, the short-chain PEG, together with HDI, contributes to the formation of the hard segment (Figure 3).

Based on the results of the swelling experiments and mechanical testing, scaffolds were fabricated from the samples PU-2, PU-3 and PU-4, which exhibited high crosslink density and sufficient mechanical strength. The scaffolds were prepared using the salt leaching technique as it will be discussed later.

### 2.3. Infrared Spectroscopy

FT-IR spectra of sunflower oil (SO), glyceride mixture (GM) and epoxidized oil (EPO) are illustrated in Figure 4.

The peaks observed at 3521 and 3463 cm^−1^ indicate the presence of hydroxyl (–OH) groups in both the GM and EPO oils. The differences in band positions suggest that the hydroxyl groups occupy different locations within the molecules. In the glyceride mixture, the –OH groups are located on the triglyceride backbone, whereas in the epoxidized oil, they are positioned along the mid-chain. This observation is consistent with ^1^H-NMR and MALDI-TOF MS results, which confirm that hydroxylation occurs concurrently with epoxidation.

The decrease in the intensity of the band around 3010 cm^−1^, along with the emergence of a signal near 830 cm^−1^ in the EPO spectrum, further supports the conversion of double bonds via epoxidation. The strong, complex band system between 2950 and 2860 cm^−1^ corresponds to the presence of numerous CH_2_ groups The characteristic vibrations of ester carbonyl (–C=O) groups are observed between 1744 and 1704 cm^−1^.

The structure of the polyurethanes containing PTMEG and oil-based polyols was also investigated using IR spectroscopy (Figure 5). In the recorded spectra, the disappearance of the broad band above 3400 cm^−1^ and the appearance of –N-H stretching vibrations at around 3337–3316 cm^−1^ indicate the formation of urethane bonds through the reaction of the polyols with HDI [29,30,31,32]. This is further supported by the absence of the isocyanate band around 2270 cm^−1^, confirming the complete consumption of isocyanate groups. The bands in the range 2942–2853 cm^−1^ belong to the C-H stretching vibration.

These observations demonstrate that the hydroxyl groups present in the oil-derived polyols exhibit sufficient reactivity to participate in urethane bond formation. A weak ester –C=O stretching band is observed between 1724 and 1721 cm^−1^, while the characteristic –C–O–C– stretching vibrations of the PTMEG units appear around 1103 cm^−1^.

### 2.4. Swelling Test

Swelling experiments provided valuable insights into the presence and characteristics of the crosslinked structure formed. During testing, the samples were immersed in toluene to induce swelling, after which key parameters such as swelling degree (Q), gel content (G), and crosslink density (ν_e_) were determined based on the measurement results.

As shown in Table 2, the swelling degree ranged from 1.4 to 2.6, while the gel content varied between 84% and 94%. The calculated crosslink densities fell within the range of 10^−3^ to 10^−4^ mol/cm^3^.

As can be surmised from the data in Table 1 and Table 2, the compositional variations among the polyurethane samples (PU-1 to PU-6) reveal a direct influence on network architecture, as reflected in the measured crosslink densities (ν_e_). The primary soft segment component is PTMEG, partially substituted in some compositions with PEG200 or PEG600, while the hard segments originate mainly from incorporation of GM and EPO reacted with 1,6-hexamethylene diisocyanate (HDI). Samples PU-3 and PU-4 have the highest HS contents (33.3% and 28.1%, respectively). These compositions yielded the highest crosslink densities (5.3 × 10^−3^ and 4.8 × 10^−3^ mol/cm^3^) accompanied by relatively low swelling degrees (Q = 1.4–1.5), indicating dense network formations. In contrast, PU-1, with one of the lowest HS content (19.3%) shows the lowest ν_e_ (7.5 × 10^−4^ mol/cm^3^), suggesting a relatively loosely connected polymer network with larger free volume. Similarly, sample PU-5, has a moderate HS fraction (25.8%) and exhibits relatively low ν_e_ (1.8 × 10^−3^ mol/cm^3^), confirming that hard segment ratio plays a dominant role in determining the swelling properties.

### 2.5. Morphology

Scanning electron microscopy (SEM) images were recorded for the prepared polyurethane samples to examine their morphology (Figure 6). The purpose of this analysis was to investigate the influence of the oil-based polyol on the polymer morphology.

The SEM images of PU-1 to PU-2 reveal a more uniform surface morphology, which can be attributed to the compatibilizing effect of the oil-based polyol. In contrast, samples PU-3 and PU-4 exhibit a characteristic grooved surface pattern, likely resulting from the presence of PEG and a higher crosslink density. Samples PU-5 and PU-6, which contain both PEG and epoxidized oil, also display an uniform surface appearance, suggesting improved phase compatibility and structural homogeneity.

### 2.6. Tensile and Relaxation Tests

The tensile and stress relaxation tests were performed to examine the influence of oil polyol and the crosslinked network on the mechanical properties of the samples. Figure 7 shows the stress vs. relative strain curves for PU 1–6.

We modeled the resulting stress–relative strain relationship using Equation (1), which accounts for multiple non-linear viscoelastic effects [33]:(1)$$\sigma (\varepsilon )=C_{1}\varepsilon^{a_{1}}+C_{2}\varepsilon^{a_{2}}\exp({-a}_{3}\varepsilon^{a_{4}})$$
where C_1_, C_2_, a_1_, a_2_, a_3_ and a_4_ are the parameters to be fitted to the experimental curves.

As shown in Figure 7, the fitted curves exhibit excellent agreement with the experimental data, and the corresponding fitting parameters are summarized in Appendix A.

As seen from the data in Appendix A, a significant deviation is observed in the values of C_2_ for samples PU-3 (8100) and PU-4 (29,500), while the rest of the samples show considerably lower C_2_ values, ranging from ~12 to ~40. This suggests that PU-3 and PU-4 possess enhanced strength and smaller elongation at break as it can also be observed by visualization of Figure 7. Furthermore, values of C_1_ shows a similar trend: while for the most samples they are between 0.15 and 0.81. However, in the case of PU-3 (6.63) and PU-4 (6.42) the values of C_1_ also stand out from the rests. Parameters a_2_ and a_3_ also indicate elevated values for PU-3 and PU-4 (notably 7.65 and 9.06 in a_3_, respectively), further supporting significant structural and/or compositional differences. In contrast, samples PU-1, PU-2, and PU-5 exhibit more similar values among parameters such as C_2_ and C_1_.

The introduction of GM leads to a substantial increase in stiffness as evident in samples PU-3 and PU-4, where Young’s modulus (E) reaches 124 MPa and 142 MPa, respectively (Table 2). However, this enhancement in stiffness comes at the expense of elasticity, with sharp elongation drop (PU-4: ε_R_ = 93%). The synergy between GM and PEG200 (PU-3) or PEG600 (PU-4) facilitates the formation of a more rigid network, confirming that GM serves as a hardening agent while PEG limits soft segment mobility.

Interestingly, EPO-based PUs maintain favorable mechanical flexibility. PU-2 exhibits an ultimate elongation of 885%, the highest among all samples, with a tensile strength of 23 MPa. This highlights the dual role of EPO in promoting elasticity and enhancing strength, likely due to the presence of reinforcing epoxide-derived crosslinks. The mechanical properties of PU-5 and PU-6 further confirm this trend, as they display a balance between high elongation (ε_R_ ≈ 680–690%) and considerable strength (σ_R_ ≈ 17.5–18.2 MPa), alongside moderate stiffness (E ≈ 22–38 MPa).

Stress relaxation experiments were also performed on PU samples 1–6, with the resulting curves shown in Figure 8a,b. As seen in Figure 8b, the relaxation curves for all PU samples can adequately be described using a stretched exponential function (Kohlrausch–Williams–Watts) Equation (2) [34].(2)$$\frac{\sigma }{\sigma_{ο}}=A\left\{\exp\left[-{\left(\frac{t}{\tau }\right)}^{\lambda }\right]-1\right\}+1$$
where A is the preexponential factor, t, and represent the time, the relaxation time and the stretching factor, respectively.

The fitted parameters of Equation (2) are listed in Appendix A, and the relaxation time distributions calculated by MATLAB (version R2024a) are also presented as Appendix A.

As illustrated in Figure 8a, the rapid decline in stress is followed by a slower relaxation process that approaches an equilibrium state. From the relative stress curves shown in Figure 8b, the samples can be ranked in the following order: PU-1 > PU-2 > PU-5 > PU-6 > PU-4 > PU-3. The relaxation behavior is strongly influenced by the crosslinked structure, allowing crosslink density (ν_r_) to be estimated from these experiments using Equation (3).(3)$$\nu_{r}=\frac{\sigma_{\mathrm{eq}}}{\mathrm{RT}\cdot \left(\lambda -\frac{1}{\lambda^{2}}\right)}$$
where λ = L_o_/L (L_o_ and L are the original and the elongated length, respectively).

The crosslink density values calculated from Equation (3) are shown in Table 3.

The crosslink density values determined by this method are in good agreement with the results obtained from the swelling experiments. Both sets of data indicate that samples PU-3, PU-4, and PU-6 possess the most highly developed crosslinked networks. In addition to crosslinking, the mechanical properties of the polymers, including their relaxation behavior, are also influenced by factors such as crystallinity and chain flexibility.

### 2.7. Thermal Properties

The DSC measurement results of PU samples 1–6 are shown in Figure 9 and Table 4.

The glass transition temperature (T_g_) of pure PTMEG is −78 °C, and as such, it was not detectable [35]. Moreover, none of the compositions showed an increase in T_g_ toward −70 °C, which is attributed to their predominantly amorphous structure. The melting temperatures (T_m_: 1–10 °C) and the calculated degrees of crystallinity (X_c_: 5–15%) of the PTMEG segments are significantly lower than those of the pure PTMEG. This reduction is due to the restricted segmental motion and limited crystallization caused by the crosslinked network. Interestingly, the PU-1 and PU-5 samples exhibit crystallinity values of 12% and 15%, respectively, corresponding to relatively lower crosslink densities, which allows for slightly greater chain mobility in these polymers.

### 2.8. Thermomechanical Properties

Figure 10 and Appendix A show the storage modulus curves obtained from DMA measurements as a function of the temperature.

The shape of these curves is governed by the stress storage capacity of the samples, which is closely linked to the crosslinked structure formed. The relatively flat regions (plateau), or more exactly slowly decreasing storage modulus values with the temperature were observed in the range of 80–120 °C for samples PU-2 and PU-4. For this case Equation (4) can be used to estimate the crosslink density (ν_D_):(4)$$\nu_{D}=\frac{E^{\prime }}{3\mathrm{RT}}$$
where E’, R and T are the storage modulus, the gas-constant and the temperature in Kelvin, respectively.

Based on the storage modulus values at 80 °C and using Equation (4), the crosslink density values for PU-2 and PU-4 were calculated to be 1.1 × 10^−3^ and 1.2 × 10^−3^ mol/cm^3^, respectively. Interestingly, a narrow plateau can also be observed on the storage modulus vs. temperature curve for sample PU-2 at the range of 40–50 °C.

### 2.9. Scaffolds from PUs 2–4

Figure 11 shows SEM images of scaffolds prepared from samples PU-2, PU-3, and PU-4. The micrographs reveal a well-developed open-cellular structure, which is a key requirement for potential applications in tissue engineering. Achieving such morphology, however, depends strongly on the material composition (with respect to ensuring biocompatibility).

The SEM images of the scaffold materials were analyzed using ImageJ software (version 1.54) [36] to determine pore size, as well as the major and minor axes of the fitted ellipses and the Feret diameter. The results of this analysis are summarized in Table 5.

Comparison of the SEM micrographs and the corresponding quantitative data reveals that the scaffold prepared from PU-3 exhibits a more extensive pore structure, which can be attributed to its higher crosslink density. In contrast, the PU-2 scaffold shows a lower pore density. As indicated by the data in Table 6, the pore sizes in all samples are smaller than the size of the salt particles used as porogens during scaffold fabrication. This discrepancy may be explained by the relatively high crosslink density, which likely results in a more compact network structure that restricts the penetration and embedding of salt crystals within the polymer matrix.

## 3. Materials and Methods

### 3.1. Materials

Sunflower seed oil purchased from COOP Hungary Ltd., Budapest was refined after pressing and/or extraction and used for the synthesis. Poly(tetramethylene ether) glycol (PTMEG2000) (M_n_ = 2 kg/mol), Poly(ethylene glycol) (PEG200) (M_n_ = 0.2 kg/mol), Poly(ethylene glycol) PEG600 (M_n_ = 0.6 kg/mol), 1,6-Hexamethylene diisocyanate (HDI) and catalyst Tin(II) 2-ethylhexanoate were obtained from Merck KGaA (Darmstadt, Germany). Toluene (analytical grade, stored on sodium wire) from Molar Chemicals Ltd. (Halásztelek, Hungary) were used.

### 3.2. Transesterification of Sunflower Oil

The transesterification of sunflower oil with glycerine was carried out the same way as detailed in Refs. [18,19,24]. Briefly, sunflower oil (5 g) and glycerol (10 g, 2:1 glycerol-to-oil ratio) were heated at 170 °C under an argon atmosphere for 6 h in the presence of a catalytic amount of calcium oxide. The resulting glyceride mixture (GM) was extracted using n-hexane.

### 3.3. Epoxidation of Sunflower Oil

To convert the unsaturated bonds of sunflower oil into epoxides [18] (EPO), 10 g of sunflower oil was dissolved in 25 mL of glacial acetic acid and heated to 65 °C under vigorous stirring in a 100 mL three-necked flask. Upon reaching the target temperature, 8.7 mL of hydrogen peroxide (30 wt %, 1.5 equivalents) was added dropwise over the course of one hour using a syringe. The reaction mixture was then maintained at 65 °C for an additional 5 h. After cooling to room temperature, the mixture was diluted with n-hexane. The organic layer was separated, washed with water five times, and dried over anhydrous MgSO_4_. Following solvent evaporation, 7.7 g of a colorless oil was obtained.

The epoxidized oil was characterized by physicochemical methods (see Figure 3), as well as by NMR spectroscopy (^1^H- and ^13^C-NMR) and MALDI-TOF mass spectrometry.

### 3.4. Synthesis of PUs

PTMEG, PEG and/or the oil-based polyol were dissolved in 30 mL of anhydrous toluene using a three-neck flask fitted with a gas inlet/outlet and a reflux condenser. The solution was stirred at 80 °C until complete dissolution was achieved. Subsequently, 1,6-hexamethylene diisocyanate (HDI) and tin(II) 2-ethylhexanoate (Sn(Oct)_2_) as a catalyst (2 mol %, 0.032 g) were introduced, and the reaction mixture was stirred for an additional 2 h at 80 °C under an argon atmosphere.

The resulting pale-yellow, viscous prepolymer was cast into a Teflon^®^ mold and allowed to cure in ambient air for 48–72 h. The resulting polymer films were subjected to characterization as described in the following section. The compositions of the reaction mixtures are summarized in Table 6.

### 3.5. Preparation of Scaffolds

The PU-2, PU-3, or PU-4 samples were prepared according to the composition described in Section 3.4, using 15 mL of anhydrous toluene. Scaffolds were then fabricated following a previously reported method [37]. During scaffold preparation, the polymer solution was poured into a Teflon^®^ dish, to which a tenfold excess of sieved NaCl crystals (200–250 µm in diameter) was added and mixed until a homogeneous dispersion was achieved.

After drying at room temperature for 3–4 days, a film approximately 5 mm in thickness was obtained. The salt was removed by repeated washing with reverse osmosis (RO) water. For subsequent testing, the samples were further dried at room temperature for several days. A flexible, open-cell, sponge-like structure was thus obtained.

### 3.6. Characterization

The acid, iodine, and hydroxyl values were determined in accordance with Hungarian standards MSZ 3633-81 [38], MSZ 3634-80 [39], and MSZ 3629 [40], or alternatively by the official ASTM D1957-86 method, respectively. The epoxy value was determined based on the MSZ EN ISO 3001:2000 [41] standard method.

Matrix-Assisted Laser Desorption/Ionization Time-of-Flight Mass Spectrometry (MALDI-TOF MS) measurements were performed using a Bruker Autoflex Speed mass spectrometer equipped with a TOF/TOF (time-of-flight/time-of-flight) mass analyzer (Bruker Daltonics, Bremen, Germany). Spectra were recorded in positive ion mode using both reflectron and linear detection. Desorption/Ionization was achieved with a solid-state laser (355 nm, ≥100 µJ/pulse) operating at 500 Hz, and spectra were obtained by summing 5000 laser shots. External calibration was carried out using a polypropylene glycol standard (M_n_ = 1000 g/mol). Samples were prepared using 2,5-dihydroxybenzoic acid (DHB) as the matrix, dissolved in tetrahydrofuran (THF) at a concentration of 20 mg/mL. The analyte and sodium trifluoroacetate (used as cationizing agent) were also dissolved in THF at concentrations of 10 mg/mL and 5 mg/mL, respectively. The components were mixed in a 10:2:1 (*v*/*v*) ratio (matrix/sample/cationizing agent), and 0.25 µL of the resulting mixture was spotted onto a metal target plate and dried in air.

^1^H-NMR and ^13^C-NMR spectra were recorded using a Bruker Avance II 500 spectrometer (500 MHz for ^1^H, 125 MHz for ^13^C) (Bruker, Karlsruhe, Germany). Deuterated chloroform (CDCl_3_) was used as the solvent, and tetramethylsilane (TMS) served as the internal standard.

Attenuated Total Reflectance Fourier Transform Infrared (ATR-FTIR) spectra were recorded using a PerkinElmer Spectrum Two FTIR spectrometer (Waltham, MA, USA). For each sample (thickness ~0.5 mm), 32 scans were averaged. Spectral data were analyzed using the Spectrum IR 2017 software package.

Scanning Electron Microscopy (SEM) was used to investigate the surface morphology of the polyurethane samples. SEM images were obtained using a Thermo Fisher Scientific Scios 2 dual-beam FIB-SEM (focused ion beam–scanning electron microscope) operated at 2 kV acceleration voltage, 50 pA beam current, and 100 ns dwell time in secondary electron (SE) mode. Prior to imaging, samples were sputter-coated with a 30 nm thick conductive gold layer.

Swelling studies were carried out by immersing polyurethane specimens (dimensions: 10 mm × 10 mm × ~0.5 mm) in 10 mL of toluene at 25 °C (298 K) in sealed containers for 24 h. The degree of swelling (Q), gel content (G), and crosslink density (ν_e_) were calculated using Equations (5)–(7) [42,43].(5)$$Q=1+\frac{\rho_{p}}{\rho_{s}}\left(\frac{m_{2}}{m_{3}}-1\right)$$(6)$$G(\%)=\frac{m_{3}}{m_{1}}\cdot 100$$
where *ρ*_s_ and *ρ*_p_ are the densities of the solvent (toluene, *ρ*: 0.8669 g/cm^3^) and the PU polymer, respectively.(7)$$\nu_{e}=\frac{-\left[\ln(1-v_{1})+v_{1}+\chi \cdot v_{1}^{2}\right]}{V_{\mathrm{ms}}\left(v_{1}^{1/3}-\frac{v_{1}}{2}\right)}$$
where ν_e_ = crosslink density, V_ms_ is the molar volume of the solvent (1.06 × 10^−4^ m^3^/mol), χ is the polymer-solvent interaction parameter (at 298 K were calculated to be 0.226), *ρ*_p_ is the density of the polymer [44].

Uniaxial tensile tests were performed using a computer-controlled INSTRON 3366 Universal Testing Machine (INSTRON, Norwood, MA, USA). In accordance with ASTM D882-12 standard [45], five dumbbell-shaped specimens (thickness ~0.5 mm, gauge length: 60 mm) were prepared from each sample and tested at a crosshead speed of 50 mm/min following appropriate conditioning. Young’s modulus (E), tensile strength, and elongation at break were determined from the resulting stress–strain curves.

Stress relaxation experiments were conducted on the same INSTRON 3366 testing system. Samples were stretched to 100% strain, and the subsequent decay in stress at constant deformation was recorded. Data were processed using Instron Bluehill Universal software (version 4.05, 2017).

Differential Scanning Calorimetry (DSC) measurements were performed using a Mettler Toledo DSC 3 instrument with power compensation (Columbus, OH, USA). To eliminate the thermal history of the polyurethane samples, an initial rapid heating was applied from −70 °C to 220 °C. Subsequent heat/cool cycles were performed under a nitrogen atmosphere. During heating, the temperature was increased from −70 °C to 220 °C at a rate of 10 °C/min. The cooling cycle followed the same rate, decreasing from 220 °C back to −70 °C. The degree of crystallinity of the PTMEG soft segment was calculated using Equation (8) [46].(8)$$C_{r}(\%)=\frac{{\Delta H}_{m}}{\chi_{\mathrm{PTMEG}}\cdot {\Delta H}_{m}^{o}}\cdot 100$$
where ΔH_m_ is the heat of fusion of the measured PU derivatives, χ is the weight fraction of PTMEG. ΔH_m_^0^ is the heat fusion of the pure 100% crystalline PTMEG [35].

Dynamic Mechanical Analysis (DMA) of the polyurethane samples was performed using a METRAVIB DMA 25 instrument (Limonest, France). Measurements were conducted in tensile mode on specimens with dimensions of 30 mm in total length (clamped length: 19 mm), 15 mm in width, and approximately 0.5 mm in thickness. A dynamic displacement amplitude of 0.1 mm was applied at a frequency of 1 Hz. The temperature was ramped from 25 °C to 150 °C at a heating rate of 2 °C/min.

## 4. Conclusions

Polyurethanes incorporating sunflower oil-derived polyols were successfully synthesized and comprehensively characterized to explore the role of bio-based soft segments in network formation and material performance. The presence of epoxidized oil polyol or glyceride mixture enabled the development of highly crosslinked structures with inherently low crystallinity. Mechanical and thermomechanical analyses confirmed favorable energy-storage capability and structural stability, particularly for PU-2 and PU-4, while SEM micrographs revealed distinctive grooved surface morphologies in GM-containing samples. Scaffolds prepared from PU-2, PU-3, and PU-4 exhibited increased bulk density and adequate mechanical integrity, indicating promising potential for biomedical use, especially in tissue engineering. Future studies may further explore composition optimization and biological evaluation to support the applicability of these materials in biomedicine.

## Data Availability

The original contributions presented in this study are included in the article/Appendix A. Further inquiries can be directed to the corresponding author.

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
