# Peer review of "Synthesis of Bio-Based Polyurethanes from Functionalized Sunflower Seed Oil"

_ijms, 2025, doi:10.3390/ijms262311380_

Round 1

Reviewer 1 Report

Comments and Suggestions for Authors

This article focuses on the synthesis of bio-based polyurethanes using sunflower seed oil-derived polyols and systematically investigates their structural, mechanical, thermal, and scaffold properties, which aligns with the sustainable development direction of polymer materials. The research design is logical, the experimental methods are relatively comprehensive, and the results effectively demonstrate the regulatory role of polyol types in material performance. I recommend this manuscript for acceptance in International Journal of Molecular Sciences after the authors address the following questions.

  1. In the Dynamic Mechanical Analysis (DMA) section (3.8), the authors observed a flat storage modulus plateau for PU-2 and PU-4 in the 80-120 °C range and calculated crosslink density using Eq. (8), but only presented data for these two samples. It is unclear why other PU samples were excluded from DMA analysis, and no explanation was provided for the narrow plateau of PU-2 in the 40-50 °C range. The authors should supplement DMA results for all samples and clarify the relationship between the plateau phenomenon and material composition to enhance the completeness of thermomechanical property analysis.
  2. The experimental results show that PU-3 and PU-4have significantly higher crosslink densities (5.3×10⁻³ and 4.8×10⁻³ mol/cm³, respectively) and Young's moduli (124 MPa and 142 MPa) than other samples, but their elongation at break (268% and 93%) is much lower. The authors attributed this to "GM serving as a hardening agent," but did not further explain why the combination of GM with PEG200/PEG600 specifically enhances rigidity while reducing elasticity. A more in-depth discussion of the interaction mechanism between GM and PEG in regulating the crosslinked network structure is needed to clarify the cause of this performance trade-off.
  3. The salt leaching method was used to fabricate porous scaffolds (2.5), and the authors chose NaCl crystals with a diameter of 200-250 μm as porogens, but did not provide a reasonable basis for this particle size selection. Additionally, the pore sizes of the prepared scaffolds (e.g., PU-2 scaffold: major diameter 48±25 μm, minor diameter 23±16 μm) are much smaller than the porogen size, and only "high crosslink density restricting salt crystal embedding" was proposed as an explanation (3.9). The authors should supplement experiments with different NaCl particle sizes to verify the effect of porogen size on scaffold pore structure and provide more direct evidence to support the proposed mechanism.
  4. In Section 3.2, the authors stated that the one-step method was chosen over the two-step method because the latter causes oil-based polyols to act as chain terminators, reducing crosslinking degree. However, no comparative experimental data (e.g., crosslink density, mechanical properties of polyurethanes synthesized via two-step method) were provided to verify this claim. Additionally, in the FT-IR analysis (3.3), the authors confirmed urethane bond formation by the disappearance of -OH peaks and appearance of -N-H peaks, but did not cite classic studies on "FT-IR characterization of polyurethane bond formation" in the field of vegetable oil-based polyurethanes, leading to insufficient depth in the argumentation. For this the authors can refer:

(1). 10.1007/s40843-025-3551-5

(2). 10.1016/j.jclepro.2018.01.193

(3). 10.1021/bm800335x

(4). 10.1016/j.carbpol.2020.117507

Author Response

                                                                                                                      November 18, 2025

Dear Reviewer,

Thank you for reviewing our manuscript entitled „Synthesis of Bio-Based Polyurethanes from Functionalized Sunflower Seed Oil”. We modified the structure and text where needed according to your comments. Our answers to your comments and questions are as follows:

This article focuses on the synthesis of bio-based polyurethanes using sunflower seed oil-derived polyols and systematically investigates their structural, mechanical, thermal, and scaffold properties, which aligns with the sustainable development direction of polymer materials. The research design is logical, the experimental methods are relatively comprehensive, and the results effectively demonstrate the regulatory role of polyol types in material performance. I recommend this manuscript for acceptance in International Journal of Molecular Sciences after the authors address the following questions.

Question: In the Dynamic Mechanical Analysis (DMA) section (3.8), the authors observed a flat storage modulus plateau for PU-2 and PU-4 in the 80-120 °C range and calculated crosslink density using Eq. (8), but only presented data for these two samples. It is unclear why other PU samples were excluded from DMA analysis, and no explanation was provided for the narrow plateau of PU-2 in the 40-50 °C range. The authors should supplement DMA results for all samples and clarify the relationship between the plateau phenomenon and material composition to enhance the completeness of thermomechanical property analysis.

Reply: Thank you for your observations. The DMA test result of sample 4 is consistent with the tensile test and stress relaxation tests. In addition to GM, the reinforcing effect of PEG is more pronounced, since its short polymer chains contribute to the formation of the crosslinked structure to a greater extent.  In the case of samples PU-1,3,5 and 6 (see Figure S6 in the Supporting Information), the storage modulus values decrease slowly as a function of temperature, and no plateau appears in the lower part of the curve. All this indicates that the crosslinked structure of the polymer is less able to store stress. Among these samples, the highest stress is required for the dynamic deformation of sample 3, which is due to the presence PEG used in addition to GM.

Question: The experimental results show that PU-3 and PU-4 have significantly higher crosslink densities (5.3×10⁻³ and 4.8×10⁻³ mol/cm³, respectively) and Young's moduli (124 MPa and 142 MPa) than other samples, but their elongation at break (268% and 93%) is much lower. The authors attributed this to "GM serving as a hardening agent," but did not further explain why the combination of GM with PEG200/PEG600 specifically enhances rigidity while reducing elasticity. A more in-depth discussion of the interaction mechanism between GM and PEG in regulating the crosslinked network structure is needed to clarify the cause of this performance trade-off.

Reply: Thank you for your notes. The synergy between GM and PEG200 (PU-3) or PEG600 (PU-4) resulted the formation of a more rigid network, based on the following recommendations:

During the preparation of the polymers, shorter chain isocyanate-terminated PEGs are formed, which enhance the crosslinking effect of GM by forming additional crosslinks by forming allophonate bonds between the linear PTMEG chains. This effect plays an important role during foaming, thus creating a sufficiently rigid structure. All this illustrated in Scheme 3.

Question: The salt leaching method was used to fabricate porous scaffolds (2.5), and the authors chose NaCl crystals with a diameter of 200-250 μm as porogens, but did not provide a reasonable basis for this particle size selection. Additionally, the pore sizes of the prepared scaffolds (e.g., PU-2 scaffold: major diameter 48±25 μm, minor diameter 23±16 μm) are much smaller than the porogen size, and only "high crosslink density restricting salt crystal embedding" was proposed as an explanation (3.9). The authors should supplement experiments with different NaCl particle sizes to verify the effect of porogen size on scaffold pore structure and provide more direct evidence to support the proposed mechanism.

Reply: Using of 200-250 mm diameter NaCl crystals as porogens was based on our previous work (Int. J. Mol. Sci. 2022, 23, 7904, (10.3390/ijms23147904)), in which we applied porogens of the same size to produce scaffolds for potential dental applications. The scaffolds thus produced were tested with dental pulp stem cells, for which the appropriate pore size is important for adhesion and growth. The pore size of the current scaffolds is smaller than that of those prepared previously, primarily due to shrinkage caused by solvent evaporation. Nevertheless, the pores remain within a suitable range to support effective cell adhesion.

Question: In Section 3.2, the authors stated that the one-step method was chosen over the two-step method because the latter causes oil-based polyols to act as chain terminators, reducing crosslinking degree. However, no comparative experimental data (e.g., crosslink density, mechanical properties of polyurethanes synthesized via two-step method) were provided to verify this claim. Additionally, in the FT-IR analysis (3.3), the authors confirmed urethane bond formation by the disappearance of -OH peaks and appearance of -N-H peaks, but did not cite classic studies on "FT-IR characterization of polyurethane bond formation" in the field of vegetable oil-based polyurethanes, leading to insufficient depth in the argumentation. For this the authors can refer:

(1). 10.1007/s40843-025-3551-5

(2). 10.1016/j.jclepro.2018.01.193

(3). 10.1021/bm800335x

(4). 10.1016/j.carbpol.2020.117507

Reply: Thank you for the suggestions. The table below presents the measurement data for the polyurethanes with similar compositions we produced in two steps:

Code

Composition

sB (MPa)

eB (%)

ne(swelling)

mol/cm3

LCS362

PTMEG-HDI-GM/4g-0,67g-1g

-*

-*

dissolved

PU-1

PTMEG-HDI-GM/4g-0,67g-1g

7.7 ± 1

429.1 ± 45

7.5 x 10-4

706

PTMEG-HDI-EPO/4g-0,67g-1g

8.1 ± 1

380 ± 42

4.5 x 10-4

PU-2

PTMEG-HDI-EPO/4g-0,67g-1g

23 ± 1.4

885 ± 34

2.3 x 10-3

*Soft film, cannot be examined

LCS362 and 706 were synthesized via a two-step process. The samples presented in the table were prepared using both one-step and two-step synthesis routes. Comparative analyses revealed significant advantages for the one-step method; consequently, the additional polyurethane compositions (PUs 3–6) were synthesized using this approach. The references you suggested have been added, as they provide further evidence supporting our conclusions.

Yours sincerely,

Sándor Kéki

        professor and head

Reviewer 2 Report

Comments and Suggestions for Authors

The manuscript details the synthesis and characterization of bio-based polyurethanes derived from functionalized sunflower seed oil. The authors prepared two types of oil polyols through transesterification and epoxidation, followed by reaction with 1,6-hexamethylene diisocyanate (HDI) and PTMEG to produce flexible and sustainable polyurethane networks. A comprehensive suite of analytical techniques (FTIR, NMR, MALDI-TOF MS, SEM, DSC, DMA, and mechanical tests) was employed to assess the structural, thermal, and mechanical properties of the resulting polymers and scaffolds. However, revisions are necessary to improve the manuscript.

The manuscript is well-structured, comprehensive, and timely, addressing the current trend toward sustainable polymer development and bio-based materials.

Comments:

Could the authors briefly quantify key findings in the abstract (e.g., tensile strength, elongation range, or crosslink density) to highlight the material performance?

Can the introduction be strengthened by comparing this system directly with similar vegetable oil–based PUs (castor, linseed, soybean,…) to emphasize novelty and performance distinction?

In scheme 2, the PU hard and soft segments are not clear. Please replace the line shape with chemical equations to show the structures of HS and SS.

In PU preparation: How the OH number of the biobased polyol is calculated and how the formulations are adjusted.

What type of polyurethane is produced, whether in the form of flexible, rigid, or film, and what are the recommended applications, should be clearly highlighted in the abstract.

In Table 1, the method for calculating HS and SS% should be clearly explained in the manuscript.

How does the ratio of EPO : GM : PTMEG specifically influence both mechanical flexibility and thermal stability?

On page 8, the Table caption is missing.

The introduction lacks the chemistry of polyurethane and explains why we need to shift to bio polyols as alternatives to petroleum-based polyols. You can use the following papers as a guide: 10.1039/D4MA01026D, and 10.1038/s41598-024-68039-w.

Research gap and the novelty of the work should be clear in the introduction.

Author Response

                                                                                                                      November 18, 2025

Dear Reviewer,

Thank you for reviewing our manuscript entitled „Synthesis of Bio-Based Polyurethanes from Functionalized Sunflower Seed Oil”. We modified the structure and text where needed according to your comments. Our answers to your comments and questions are as follows:

The manuscript details the synthesis and characterization of bio-based polyurethanes derived from functionalized sunflower seed oil. The authors prepared two types of oil polyols through transesterification and epoxidation, followed by reaction with 1,6-hexamethylene diisocyanate (HDI) and PTMEG to produce flexible and sustainable polyurethane networks. A comprehensive suite of analytical techniques (FTIR, NMR, MALDI-TOF MS, SEM, DSC, DMA, and mechanical tests) was employed to assess the structural, thermal, and mechanical properties of the resulting polymers and scaffolds. However, revisions are necessary to improve the manuscript.

Comment: The manuscript is well-structured, comprehensive, and timely, addressing the current trend toward sustainable polymer development and bio-based materials.

Reply: Thank you for the kind words.

Comment: Could the authors briefly quantify key findings in the abstract (e.g., tensile strength, elongation range, or crosslink density) to highlight the material performance?

Reply: Thanks for my reviewer for his suggestion. The abstract modified as follows:

„The resulting PU films exhibited excellent flexibility (885 %) and mechanical properties (23 MPa), as evaluated by ATR-FTIR, Tensile test, DSC, DMA and SEM methods. The crosslink density of the order of 10-3 also contributes to the development of outstanding mechanical properties.”

Comment: Can the introduction be strengthened by comparing this system directly with similar vegetable oil–based PUs (castor, linseed, soybean,…) to emphasize novelty and performance distinction?

Reply: Thanks for my reviewer for his suggestion. The introduction modified as follows:

„This polyurethane system integrates the advantageous properties of the vegetable oil–based polyol and PTMEG, achieving a balance between mechanical strength and elasticity. In contrast, polyurethanes synthesized solely from simple vegetable oils (such as castor, linseed, or soybean oil) and diisocyanates (isophorone diisocyanate, IPDI, or methylene diphenyl diisocyanate, MDI) exhibit significantly lower elasticity.5,22

Comment: In scheme 2, the PU hard and soft segments are not clear. Please replace the line shape with chemical equations to show the structures of HS and SS.

Reply: Thank you for your note, Scheme 2 was modified accordingly. Furthermore, the article has been supplemented with Scheme 3.

Comment: In PU preparation: How the OH number of the biobased polyol is calculated and how the formulations are adjusted.

Reply: Thank you for your note. The OH numbers of the oil polyols were determined according to the following formula:

where

56.1 = KOH equivalent weight

N = Nominal normality of KOH solution multiplied by a factor

V2 = Volume of KOH solution consumed during titration of blank sample (cm3)

V1 = Volume of KOH solution consumed during titration of sample (cm3)

a = mass of the sample (g)

S = Acid value of the tested oil

given in Hungarian patent No MSZ 3629. This calculation procedure was inserted into the Supporing Information.

Comment: What type of polyurethane is produced, whether in the form of flexible, rigid, or film, and what are the recommended applications, should be clearly highlighted in the abstract.

Reply: Thank you for the suggestion. The following sentence was inserted into abstract:

„Given their composition and favorable physico-chemical properties, these materials may represent promising candidates for the design and development of advanced biomedical systems.”

Comment: In Table 1, the method for calculating HS and SS% should be clearly explained in the manuscript.

Reply: Thank you for your suggestion. The HS and SS amounts were calculated based on the masses given in Table 6. PTMEG and oil polyol were taken as soft segments, while HDI, which plays the role of a linker, was considered as hard segments, in addition to the shorter chain PEG 200 and 600. Which are indicated now in the caption of the table.

„HS and SS content calculated from the masses: HDI, PEG denoted as HS, while PTMEG and oil polyol confirm as SS.”

Comment: How does the ratio of EPO : GM : PTMEG specifically influence both mechanical flexibility and thermal stability?

Reply: Thank you for this comment. In PU-1 and PU-2, only PTMEG - GM and PTMEG - EPO are present in equal proportions. In samples PUs 3-6, PTMEG is present in a smaller proportion compared to the previous samples, which is reflected in the decrease in flexibility. This is higher in the case of PTMEG-GM-PEG (PU-3 and 4).

We can still draw conclusions about the thermal stability of the samples based on the DMA measurements. TGA measurements were not performed. In the case of the PU-2 sample, the epoxidized oil provides greater stability to the sample by creating a higher crosslink density, while the more beneficial effect of PEG is shown next to GM (see PU-3 and 4).

Comment: On page 8, the Table caption is missing.

Reply: The mentioned table on page 5. is given as a part of Figure 3.

Comment: The introduction lacks the chemistry of polyurethane and explains why we need to shift to bio polyols as alternatives to petroleum-based polyols. You can use the following papers as a guide: 10.1039/D4MA01026D, and 10.1038/s41598-024-68039-w.Research gap and the novelty of the work should be clear in the introduction.

Reply: Thank you for your notes. The introduction modified as follows and the recommended articles cited as refs 1 and 3.

„Polyurethanes are synthesized from polyols and isocyanates through the formation of urethane linkages. The relative ratio and molecular structure of these components determine the properties of the resulting polymer.¹ Traditionally, polyurethane production has relied predominantly on petroleum-based raw materials, thereby contributing to increased anthropogenic environmental impact. In recent years, however, the incorporation of bio-based polyols into polyurethane systems² has attracted considerable attention as a sustainable alternative, driven by growing environmental concerns associated with conventional plastics.3”

Yours sincerely,

Sándor Kéki

        professor and head

Round 2

Reviewer 1 Report

Comments and Suggestions for Authors

good job

Author Response

Thank you very much for reviewing our manuscript.

Reviewer 2 Report

Comments and Suggestions for Authors

The authors have addressed all my concerns, and the paper is now ready for publication in its current form.

Author Response

(The authors gave the same response as above.)
